# WHY SAM FINETUNING CAN BENEFIT OOD DETECTION?

## ABSTRACT

The out-of-distribution (OOD) detection task is crucial for the real-world deployment of machine learning models. In this paper, we propose to study the problem from the perspective of Sharpness-aware Minimization (SAM). Compared with traditional optimizers such as SGD, SAM can better improve the model performance and generalization ability, and this is closely related to OOD detection (Vaze et al., 2021). Therefore, instead of using SGD, we propose to fine-tune the model with SAM, and observe that the score distributions of in-distribution (ID) data and OOD data are pushed away from each other. Besides, with our carefully designed loss, the fine-tuning process is very time-efficient. The OOD performance improvement can be observed after fine-tuning the model within 1 epoch. Moreover, our method is very flexible and can be used to improve the performance of different OOD detection methods. The extensive experiments have demonstrated that our method achieves *state-of-the-art* performance on widely-used OOD benchmarks across different architectures. Comprehensive ablation studies and theoretical analyses are discussed to support the empirical results.

## 1 INTRODUCTION

Deep learning models are known to be fragile to distribution shifts. Detecting Out-of-Distribution (OOD) data is thus important for the real-life deployment of machine learning models (Song et al., 2022). The major difficulty arises from the fact that contemporary deep learning models can make excessively confident but incorrect predictions for OOD data (Nguyen et al., 2015). To address this challenge, previous *post hoc* methods distinguish in-distribution (ID) and OOD data either by using feature distance (Lee et al., 2018b), gradient abnormality (Huang et al., 2021), or logit manipulation (Liu et al., 2020). Although these *post hoc* methods do not need to fine-tune or retrain the models, the performances are usually inferior compared with the supervised ones which require further training/fine-tuning. Also, *post hoc* methods might be confined to certain network architectures and benchmarks.

From the perspective of the availability of extra training/fine-tuning data, the supervised methods can be divided into three types: (1) no extra data; (2) using real OOD data; (3) using pseudo-OOD data (generated from ID data). For the first type of methods, Huang & Li (2021) proposed MOS to fine-tune the pretrained model. By reducing the number of in-distribution classes through grouping similar semantic categories, the model can simplify the decision boundary and reduce the uncertainty space between ID and OOD data. However, this method requires additional human labor to group similar semantic categories and select the suitable number of groups. The KL Matching (Hendrycks et al., 2022) method also belongs to the first type. It learns a new distribution for each class from the validation ID dataset to enhance multi-class OOD detection. However, these methods usually have inferior performance compared with the other two types of methods which use extra OOD data.

For instance, as a representative of the second type of methods, OE (Hendrycks et al., 2018) requires additional training with extra real OOD data. However, it has quite a few inevitable drawbacks, such as its reliance on a large amount of real OOD data (*i.e.* real OOD images rather than pseudo ones), costly training time which usually takes many epochs (>5), and poor robustness across different network architectures and benchmarks. These drawbacks limit the practical application of the method, as well as other methods which use real OOD data.

To avoid using real OOD data, the third type of methods has been proposed which uses pseudo-OOD data. These methods are more user-friendly and have practical applications. Usually, they have better performance than the first type. Taking the FeatureNorm (Yu et al., 2023) method for example, it uses the pseudo-OOD data to select the block with best OOD detection ability, then computes the feature norm of the selected block as OOD detection score. However, its performance is slightly inferior compared with the second type of methods which uses real OOD data. Because of its practical application, this paper focuses on the third type of methods which are fine-tuned with pseudo-OOD data.

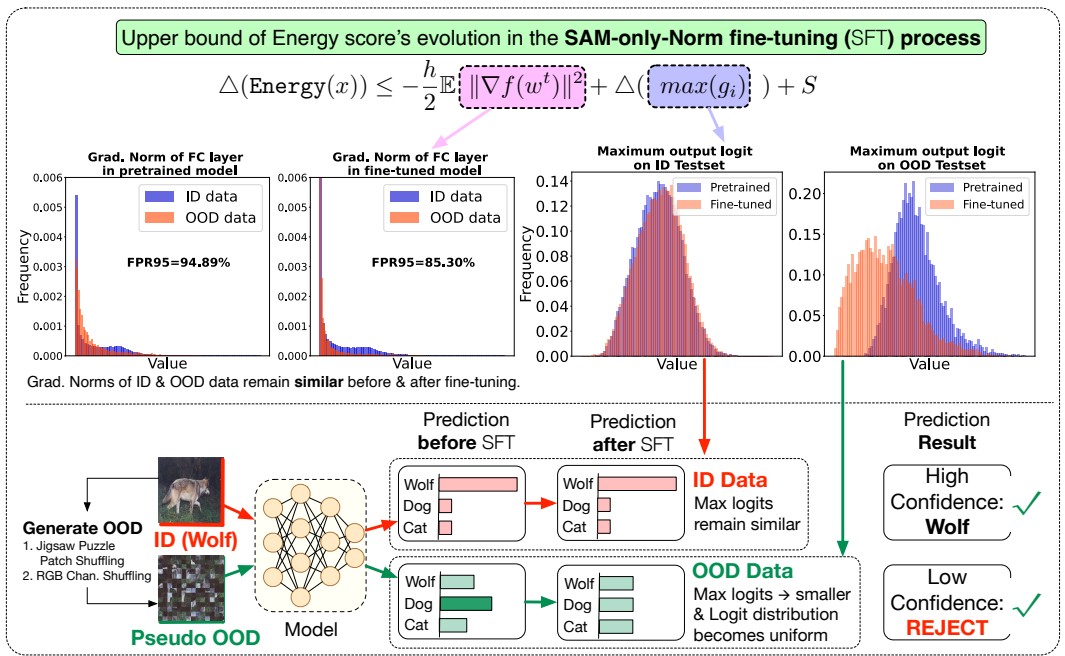

Figure 1: Our `SFT` fine-tunes the pretrained model using ID and pseudo-OOD data. In the fine-tuning process, SAM incorporates two additional terms which can reflect the upper bound of the change of `Energy` score: the gradient norm $\|\nabla f(w^t)\|^2$ and the maximum output logit $max(g_i)$. As can be seen in the distribution of these two terms, the gradient norm distributions of ID and OOD data $\|\nabla f(w^t)\|^2$ remain similar before and after fine-tuning, while the maximum output logit distribution of OOD data $max(g_i)$ decreases more than that of ID data, making the OOD score distribution flat and move more to the left. As demonstrated by the example at the bottom, the upper bound change finally leads to better score distributions for OOD and eventually helps to better distinguish the ID and OOD samples.

Recently, Sharpness-aware Minimization (SAM) indicates a novel training framework by simultaneously minimizing the loss value and the sharpness. The concept of sharpness defines the smoothness of the data point in the neighborhood of the loss landscape. *Intuitively, the network is well-trained on ID samples, and the loss landscape might be flat and smooth. On the other hand, the OOD samples which the model has never seen before are more likely to lie on a sharp and noisy landscape.* We argue this clue could be leveraged for distinguishing ID and OOD samples. **We propose our Sharpness-aware Fine-Tuning (`SFT`) method, in which the pretrained model is fine-tuned by SAM using pseudo-OOD data within 1 epoch (50 steps more precisely).** Fig. 1 depicts the overall procedure of the fine-tuning process. Our `SFT` generates pseudo-OOD data using both *Jigsaw Puzzle Patch Shuffling* and *RGB Channel Shuffling*. As can be seen in Fig. 2, ID data and OOD data are better distinguished. Our `SFT` has improved the OOD detection performance of typical baseline OOD methods (*i.e.,* Energy (Liu et al., 2020), MSP (Hendrycks & Gimpel, 2017) and RankFeat (Song et al., 2022)) by a large margin, *i.e.*, $17.60\%$ in FPR95 and $5.10\%$ in AUROC on average. In particular, for some *post-hoc* methods which are originally sensitive to specific architectures, our fine-tuning method can alleviate such a problem and improve their performance to a competitive level. For instance, GradNorm (Huang et al., 2021) which originally has a $92.68\%$ FPR95 on Textures can achieve a $51.03\%$ FPR95 after fine-tuning. Extensive ablation studies are

performed to reveal important insights of `SFT`, and concrete theoretical analysis is conducted to explain the underlying mechanism and support our empirical observation.

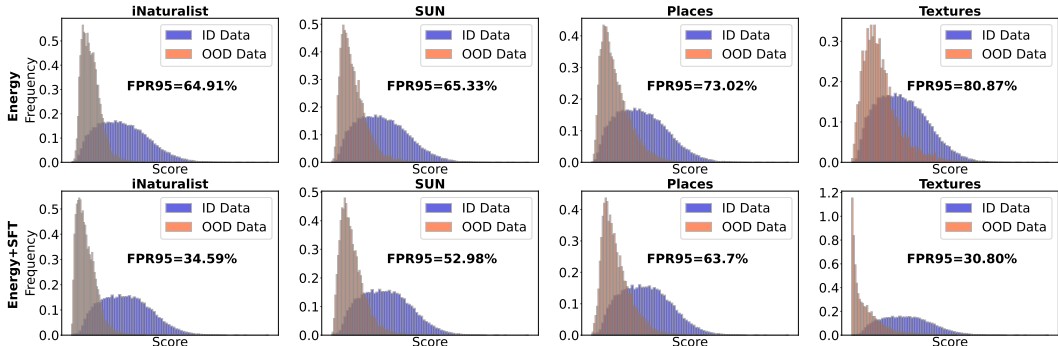

Figure 2: The score distributions of `Energy` (Liu et al., 2020) (*top row*) and our proposed `SFT+Energy` (*bottom row*) on four OOD datasets. Our `SFT` helps distinguish ID and OOD data.

The key results and main contributions are threefold:

- We propose Sharpness-aware Fine-Tuning (`SFT`), a simple, real OOD data-free, and time-efficient method for OOD detection by fine-tuning pre-trained model with SAM using generated pseudo-OOD data within 1 epoch. Comprehensive experimental results indicate that our `SFT` brings large improvements in various OOD detection benchmarks and methods.

- We perform extensive ablation studies on the impact of (1) fine-tuning epochs; (2) the effectiveness of SAM-only norm; (3) the perturbation radius $\rho$ in SAM; (4) the advantage of SAM over other optimizers. These results carefully motivate how we design our method and tune the hyper-parameters.

- Theoretical analyses are conducted to shed light on the working mechanism: by visualizing the two vital terms in the upper bound of the `Energy` score change: the distribution of the gradient norm and the change of the model's max output logit, we found that the latter one effectively encourages the separation of the ID and OOD data while the relative distribution difference between the ID and OOD data reflected by the first term stays the same.

Besides, similar observations could be obtained for other OOD detection architectures on different testsets. For details, please refer to Appendix A.2.

## 2 RELATED WORK

**Sharpness-aware Minimization.** Traditional optimizers are unable to search for global optima that is smooth enough in the neighborhood. To overcome this drawback, Sharpness-aware Minimization (SAM) is proposed to simultaneously minimize the loss value and the loss sharpness in the landscape (Foret et al., 2020). Over the years, SAM has been proven to improve the generalization ability of deep models under various settings (Bahri et al., 2022; Chen et al., 2021; Qu et al., 2022). There are several variants of the original SAM which further improve the optimization, such as ASAM (Kwon et al., 2021), GSAM (Zhuang et al., 2021), SAM-only-norm (Mueller et al., 2023), and SSAM (Mi et al., 2022). Another line of research tackles the problem of increased computational cost and proposes different acceleration techniques, including ESAM (Du et al., 2021) and LookSAM (Liu et al., 2022b). Last but not least, Bartlett et al. (2022) and Andriushchenko & Flammarion (2022) aim to elucidate the working mechanism of SAM from different perspectives.

**Distribution shift.** Distribution shifts have been a persistent challenge in the field of machine learning research (Wiles et al., 2022; Quiñonero-Candela et al., 2008; Hand, 2006; Koh et al., 2021). The problem of distribution shifts can be broadly classified into two categories: shifts in the input space and shifts in the label space. When the shifts occur solely in the input space, they are commonly referred to as covariate shifts (Hendrycks & Dietterich, 2019). In this setting, the input data is subject to perturbations or domain shifts, while the label space remains unchanged (Ovadia et al.,

2019). The primary goal is to improve the robustness and generalization of a model (Hendrycks et al., 2019). In the task of OOD detection, the labels are mutually exclusive, and the primary focus is to determine whether a test sample should be predicted by the pre-trained model (Hsu et al., 2020).

**OOD detection with discriminative models.** The initial research on discriminative OOD detection can be traced back to the classification models that incorporate a rejection option (Fumera & Roli, 2002). One major direction of OOD detection is to design effective scoring functions, mainly consisting of confidence-based methods (Hendrycks & Gimpel, 2017; Liang et al., 2018; Bendale & Boult, 2016), distance-based approaches (Lee et al., 2018b; Sun et al., 2022; Ren et al., 2022; Sehwag et al., 2020), energy-based score (Liu et al., 2020; Sun et al., 2021; Wu et al., 2022; Song et al., 2022), gradient-based score (Huang et al., 2021), and Bayesian methods (Gal & Ghahramani, 2016; Malinin & Gales, 2019; Lakshminarayanan et al., 2017; Wen et al., 2019). Another promising line of OOD detection is to regularize the pre-trained model to produce lower confidence on OOD data (Lee et al., 2018a; Hendrycks et al., 2018). Compared with *post hoc* scoring functions, these approaches usually achieve superior OOD detection performance. However, the above methods often require extra real OOD data and long training time, which limits their deployment in real-world applications. In contrast, our SFT is both OOD-data-free and time-efficient, which can be successfully applied to various OOD detection baselines.

**OOD detection with generative models.** In contrast to discriminative models, generative models detect OOD samples by estimating the probability density function (Kingma & Welling, 2014; Huang et al., 2017; Rezende et al., 2014; Van den Oord et al., 2016). A sample with a low probability is considered OOD data. In recent years, numerous approaches have leveraged generative models for the purpose of OOD detection (Ren et al., 2019; Serrà et al., 2019; Wang et al., 2020; Xiao et al., 2020; Schirrmeister et al., 2020; Kirichenko et al., 2020). Nonetheless, as highlighted in Nalisnick et al. (2019), it has been noted that generative models have the risk of assigning high likelihood values to OOD data. Recent trends in generative models try to enhance the OOD detection performance by utilizing synthesizing outliers as regularization. Graham et al. (2023); Liu et al. (2023) utilized diffusion models for OOD detection, where the reconstruction error was applied as the OOD score. Liu et al. (2022a) concurrently trained a small-scale diffusion model and a classifier, considering the interpolation between ID data and its noisy variants as outliers.

## 3 METHODOLOGY

In this section, we first introduce the OOD detection task. Then we present our proposed SFT that conducts OOD detection by fine-tuning with SAM.

**Preliminary: OOD detection task.** The OOD detection problem usually is defined as a binary classification problem between ID and OOD data. For a given model $f$ which is trained on ID data $\mathcal{D}_{id}$, OOD detection aim to design the score function $\mathcal{G}$:

$$\mathcal{G}(\mathbf{x}) = \begin{cases} \text{in} & \mathcal{S}(\mathbf{x}) > \gamma, \\ \text{out} & \mathcal{S}(\mathbf{x}) < \gamma. \end{cases} \tag{1}$$

where $x$ denotes the sample to be tested, $\mathcal{S}(.)$ is the seeking score function. $\gamma$ is the chosen threshold to distinguish whether the given sample is ID or OOD. The most crucial challenge in OOD detection is how to design an effective score function to distinguish between the two types of data.

**Sharpness-aware Fine-Tuning (SFT).** The SAM optimizer is proposed to minimize the loss value and the loss sharpness in the optimization landscape. Recently, it has been proven to be applicable to multiple machine learning tasks (Foret et al., 2020). Motivated by OE (Hendrycks et al., 2018), we fine-tune the pre-trained model using SAM to push the ID and OOD score distributions away. Similar to OE, given the data distributions $\mathcal{D}_{in}$ and $\mathcal{D}_{out}$, we fine-tune the pre-trained model $f$ using the loss function $L_{\text{SFT}}$ defined as:

$$L_{\text{SFT}} = E_{(x,y) \sim \mathcal{D}_{in}} L_{CE}(f(x), y) + \lambda E_{x' \sim \mathcal{D}_{out}} D(f(x'), U) \tag{2}$$

where $L_{CE}(f(x), y)$ is the cross-entropy loss. The key to improving the OOD detection performance is to minimize $D(f(x'), U)$ which measures the statistical distance between the output of OOD samples and the uniform distribution. To craft $\mathcal{D}_{out}$, we use both *Jigsaw Puzzle Patch Shuffling* and *RGB Channel Shuffling* methods to generate pseudo-OOD data. More specifically, *Jigsaw*

*Puzzle Patch Shuffling* divides the original ID image into small patches and then reassembles these patches randomly to generate new images, while *RGB Channel Shuffling* randomly shuffles the three RGB channels of the image. At each fine-tuning iteration, we use both *Jigsaw Puzzle Patch Shuffling* and *RGB Channel Shuffling* to generate pseudo-OOD data. Some examples are visualized in Appendix A.2.

The original SAM enforces perturbations to all the parameters of the model. This may introduce strong interference and negatively influence the OOD detection performance. We give a preliminary analysis of this phenomenon in the ablation study of Sec. 4.3. Therefore, we choose SAM-only-norm (Mueller et al., 2023), an improved version of SAM which only perturbs the normalization layers, as our optimizer throughout the experiments and theoretical analysis.

**Theoretical Analysis.** In this part, we first present the upper bound of the change of `Energy` score at each fine-tuning iteration. Then we illustrate the changes of the upper bound formula by analyzing the specific terms through numerical experiments. From the perspective of the upper bound, we explain why `SFT` could improve the OOD performance of the model.

**Proposition 1.** *The upper bound of Energy score's change during the SAM-only-norm fine-tuning process is defined as:*

$$\triangle(Energy(x)) \leq -\frac{h}{2}\mathbb{E}\|\nabla f(w^t)\|^2 + \triangle(max(g_i)) + S \tag{3}$$

*where $g(w^t) = [g_1, g_2, ..., g_n]$ denotes the logit output of the model, $\nabla f(w^t)$ is the gradient of the loss function $f(w^t)$, $w^t$ denotes the parameters of model, $h$ represents the learning rate and $S$ is an irreducible constant. The upper bound of the change of Energy score can effectively improve the separation between ID and OOD data.*

For the detailed derivations of the upper bound, please refer to the Appendix. We start by analyzing the *r.h.s* terms of Eq. (3), *i.e.,* the gradient norm, and the max logit. Fig. 3 presents the distribution of gradient norm and the max logit of the ID and OOD data. We can observe that gradient norm distributions are only mildly influenced as shown in Fig. 3 (a), but the OOD data would decrease more in the max logit, making the distribution of the OOD scores decrease faster (within 50 training steps/mini-batches) and move more to the left as shown in Fig. 3 (b). This would help the model to distinguish the ID and OOD samples. Fig. 4 displays the score distributions of the `Energy` method. The empirical observation meets our analysis of Prop. 3. We notice that similar analysis can be extended to MaxLogit and other energy-based score, *i.e.,* RankFeat (Song et al., 2022) and ReAct (Sun et al., 2021).

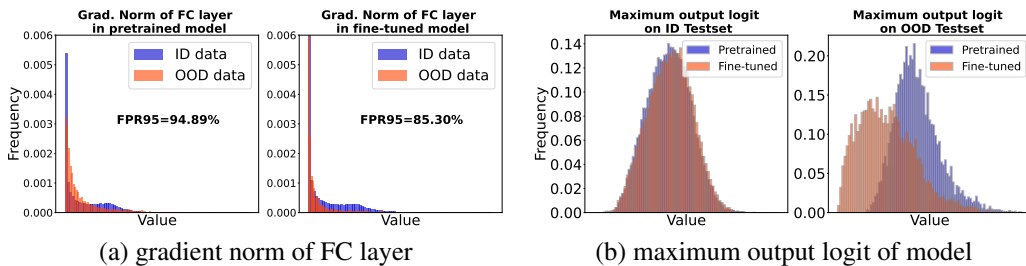

(a) gradient norm of FC layer      (b) maximum output logit of model

Figure 3: (a) The gradient norm of FC layer in the pretrained and fine-tuned model; (b) maximum output logit of model on ID and OOD testset. The above results are reported on ResNetv2-101 with ImageNet-1k (Deng et al., 2009) as ID set and Textures (Cimpoi et al., 2014) as OOD set.

# 4 EXPERIMENTS

## 4.1 SETUP

**Dataset.** In line with Huang & Li (2021), we use the large-scale ImageNet-1k benchmark (Deng et al., 2009) to evaluate our method, which is more challenging than the traditional benchmarks. We use four test sets as our OOD datasets, namely iNaturalist (Van Horn et al., 2018), SUN (Xiao et al., 2010), Places (Zhou et al., 2017), and Textures (Cimpoi et al., 2014). Besides the experiment

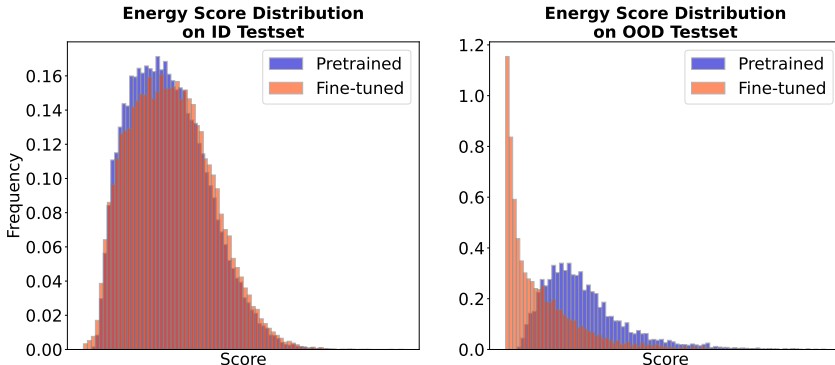

Figure 4: The score distributions of ID and OOD data using `Energy` (Liu et al., 2020) evaluated on ResNetv2-101 (He et al., 2016b) with ImageNet-1k (Deng et al., 2009) set as ID set and Textures (Cimpoi et al., 2014) as OOD set. The ID scores remain relatively unchanged while OOD scores decrease a lot.

on the large-scale benchmark, we also evaluate our method on CIFAR (Krizhevsky et al., 2009) benchmark where the OOD datasets consist of SVHN (Netzer et al., 2011), Textures (Cimpoi et al., 2014), LSUN-crop (Yu et al., 2015), LSUN-resize (Yu et al., 2015), iSUN (Xu et al., 2015) and Places365 (Zhou et al., 2017).

**Models.** Following Huang et al. (2021), the main evaluation is done using Google BiT-S model (Kolesnikov et al., 2020) pretrained on ImageNet-1k with ResNetv2-101 (He et al., 2016b). We also evaluate our method with T2T-ViT-24 (Yuan et al., 2021), which is a tokens-to-tokens vision transformer that exhibits competitive performance against CNNs when trained from scratch.

**Metrics.** We mainly use FPR95 and AUROC as the evaluation metrics. FPR95 refers to the probability of misclassifying samples when the true positive rate (TPR) reaches 95%. Lower FPR95 implies less overlapping between ID and OOD scores. AUROC denotes the area under the receiver operating characteristic curve (ROC). Higher AUROC indicates better performance.

**Implementation Details.** Throughout all the experiments, we set the OOD loss strength $\lambda$ of Eq. (2) as 1. For the pseudo-OOD data generated from *Jigsaw Puzzle Patch Shuffling*, we randomly select the patch size from $[2, 3, 5, 10, 15, 30, 60]$ at every iteration. Our fine-tuning lasts only for 50 steps with a learning rate of $1e{-}3$ and weight decay of $1e{-}5$. The perturbation radius $\rho$ of SAM is set to 0.5. At the inference time, all images are resized to the resolution of $480 \times 480$.

## 4.2 RESULTS

Table 1: Main evaluation results on ResNetv2-101 (He et al., 2016b). All values are reported in percentages, and the best three results are highlighted with **red**, **blue**, and **cyan**.

| Methods | iNaturalist | | SUN | | Places | | Textures | | Average | |
|---|---|---|---|---|---|---|---|---|---|---|
| | FPR95 ($\downarrow$) | AUROC ($\uparrow$) | FPR95 ($\downarrow$) | AUROC ($\uparrow$) | FPR95 ($\downarrow$) | AUROC ($\uparrow$) | FPR95 ($\downarrow$) | AUROC ($\uparrow$) | FPR95 ($\downarrow$) | AUROC ($\uparrow$) |
| **MSP** (Hendrycks & Gimpel, 2017) | 63.69 | 87.59 | 79.89 | 78.34 | 81.44 | 76.76 | 82.73 | 74.45 | 76.96 | 79.29 |
| ODIN (Liang et al., 2018) | 62.69 | 89.36 | 71.67 | 83.92 | 76.27 | 80.67 | 81.31 | 76.30 | 72.99 | 82.56 |
| **Energy** (Liu et al., 2020) | 64.91 | 88.48 | 65.33 | 85.32 | 73.02 | 81.37 | 80.87 | 75.79 | 71.03 | 82.74 |
| Mahalanobis (Lee et al., 2018b) | 96.34 | 46.33 | 88.43 | 65.20 | 89.75 | 64.46 | 52.23 | 72.10 | 81.69 | 62.02 |
| GradNorm (Huang et al., 2021) | 50.03 | 90.33 | 46.48 | 89.03 | 60.86 | 84.82 | 61.42 | 81.07 | 54.70 | 86.71 |
| ReAct (Sun et al., 2021) | 44.52 | 91.81 | 52.71 | 90.16 | 62.66 | 87.83 | 70.73 | 76.85 | 57.66 | 86.67 |
| **RankFeat** (Song et al., 2022) | 41.31 | 91.91 | 29.27 | 94.07 | 39.34 | 90.93 | 37.29 | 91.70 | 36.80 | 92.15 |
| KL Matching (Hendrycks et al., 2022) | 27.36 | 93.00 | 67.52 | 78.72 | 72.61 | 76.49 | 49.70 | 87.07 | 54.30 | 83.82 |
| MOS (Huang & Li, 2021) | 9.28 | 98.15 | 40.63 | 92.01 | 49.54 | 89.06 | 60.43 | 81.23 | 39.97 | 90.11 |
| **MSP+SFT** | 36.59 | 92.81 | 67.32 | 83.05 | 73.2 | 80.40 | 38.72 | 89.34 | 53.96 | 86.4 |
| **Energy+SFT** | 34.59 | 94.21 | 52.98 | 89.23 | 63.7 | 85.04 | 30.80 | 91.94 | 45.52 | 90.11 |
| **RankFeat+SFT** | 37.15 | 92.93 | 27.66 | 94.35 | 37.88 | 91.14 | 27.29 | 93.83 | 32.50 | 93.06 |

**Main Results.** Table 1 compares the performance of some baseline methods and our fine-tuning method. We combine SFT with several *post hoc* methods and observe that our SFT can improve the performance of these *post hoc* methods by a large margin.

- Specifically, for MSP (Hendrycks & Gimpel, 2017), SFT improves the baseline by 23% in the average FPR95 and 7.11% in the average AUROC.

- For Energy (Liu et al., 2020), SFT boosts the performance by 25.51% in the average FPR95 and 7.37% in the average AUROC.

- Equipped with our SFT, RankFeat (Song et al., 2022) achieves the performance gain of 4.3% in the average FPR95 and 0.91% in the average AUROC.

- Compared to Huang & Li (2021); Hendrycks et al. (2022) that require a large amount of training time, our approach has better performance while taking much less training time.

Table 2: Evaluation results on T2T-ViT-24 (Yuan et al., 2021). All values are reported in percentages, and the best two results are highlighted with **red** and **blue**.

| Model | Methods | iNaturalist FPR95 (↓) | iNaturalist AUROC (↑) | SUN FPR95 (↓) | SUN AUROC (↑) | Places FPR95 (↓) | Places AUROC (↑) | Textures FPR95 (↓) | Textures AUROC (↑) | Average FPR95 (↓) | Average AUROC (↑) |
|---|---|---|---|---|---|---|---|---|---|---|---|
| **T2T-ViT-24** | **MSP** Hendrycks & Gimpel (2017) | 48.92 | 88.95 | 61.77 | 81.37 | 69.54 | 80.03 | 62.91 | 82.31 | 60.79 | 83.17 |
| | ODIN Liang et al. (2018) | 44.07 | 88.17 | 63.83 | 78.46 | 68.19 | 75.33 | 54.27 | 83.63 | 57.59 | 81.40 |
| | Mahalanobis Lee et al. (2018b) | 90.50 | 58.13 | 91.71 | 50.52 | 93.32 | 49.60 | 80.67 | 64.06 | 89.05 | 55.58 |
| | **Energy** (Liu et al., 2020) | 52.95 | 82.93 | 68.55 | 73.06 | 74.24 | 68.17 | 51.05 | 83.25 | 61.70 | 76.85 |
| | GradNorm (Huang et al., 2021) | 99.30 | 25.86 | 98.37 | 28.06 | 99.01 | 25.71 | 92.68 | 38.80 | 97.34 | 29.61 |
| | ReAct (Sun et al., 2021) | 52.17 | 89.51 | 65.23 | 81.03 | 68.93 | 78.20 | 52.54 | 85.46 | 59.72 | 83.55 |
| | **RankFeat** (Song et al., 2022) | 50.27 | 87.81 | 57.18 | **84.33** | 66.22 | 80.89 | 32.64 | 89.36 | 51.58 | 85.60 |
| | **MSP+SFT** | 41.98 | 90.97 | 65.41 | 82.88 | 68.03 | **81.99** | 47.02 | 88.12 | 55.61 | 85.99 |
| | **Energy+SFT** | **30.90** | **92.13** | **56.02** | 83.27 | **63.29** | 79.60 | **22.27** | **94.56** | **43.12** | **87.39** |
| | **RankFeat+SFT** | **30.62** | **93.23** | **49.88** | **88.37** | **59.29** | **85.52** | **18.26** | **93.20** | **39.51** | **90.08** |

**Our SFT also suits architectures of different normalization layers.** Since our SAM-only-norm optimizer perturbs only the normalization layer, it is not clear whether the performance gain is consistent with other normalization layers. To investigate this question, we evaluate our method on T2T-ViT-24 (Yuan et al., 2021) which uses LayerNorm (Ba et al., 2016) throughout the network. Table 2 compares the performance of some baseline methods as well as our SFT. For MSP, Energy, and RankFeat, our method brings the average performance gain of 9.10% in FPR95 and 5.95% in AUORC. The consistent improvement across methods demonstrates that our fine-tuning suits networks with different normalization layers.

**Our SFT is also applicable to other benchmarks.** Now we turn to evaluating our approach on CIFAR-10 benchmark with VGG11 (Simonyan & Zisserman, 2015). Table 3 compares our method with some *post hoc* methods and training-needed methods. Similar to the ImageNet-1K benchmark, our SFT improves the Energy baseline by 19.59% in the average FPR95 and 3.53% in the average AUROC. Moreover, our SFT can boost the performance of the simple MSP to a very competitive level: 35.05% in the average FPR95 and 93.92% in the average AUROC.

Table 3: Evaluation results on VGG11 (Simonyan & Zisserman, 2015). All values are reported in percentages, and the best two results are highlighted with **red** and **blue**.

| Model | Methods | SVHN FPR95 (↓) | SVHN AUROC (↑) | Textures FPR95 (↓) | Textures AUROC (↑) | LSUN-crop FPR95 (↓) | LSUN-crop AUROC (↑) | LSUN-resize FPR95 (↓) | LSUN-resize AUROC (↑) | iSUN FPR95 (↓) | iSUN AUROC (↑) | Places365 FPR95 (↓) | Places365 AUROC (↑) | Average FPR95 (↓) | Average AUROC (↑) |
|---|---|---|---|---|---|---|---|---|---|---|---|---|---|---|---|
| **VGG11** | **MSP** Hendrycks & Gimpel (2017) | 68.07 | 90.02 | 63.86 | 89.37 | 46.63 | 93.73 | 70.19 | 86.29 | 71.81 | 85.71 | 68.08 | 87.25 | 64.77 | 88.73 |
| | ODIN Liang et al. (2018) | 53.84 | 92.23 | 48.09 | 91.94 | 19.95 | 97.01 | 54.29 | 89.47 | 56.61 | 88.87 | 52.34 | 89.86 | 47.52 | 91.56 |
| | **Energy** (Liu et al., 2020) | 53.13 | 92.26 | 47.04 | 92.08 | 18.51 | 97.20 | 53.02 | 89.58 | 55.39 | 88.97 | 51.67 | **89.95** | 46.46 | 91.67 |
| | ReAct (Sun et al., 2021) | 58.81 | 83.28 | 51.73 | 87.47 | 23.40 | 94.77 | 47.19 | 89.68 | 51.30 | 88.07 | **50.47** | 87.39 | 47.15 | 88.44 |
| | DICE (Sun & Li, 2022) | 47.81 | **93.27** | 50.95 | 91.77 | 16.73 | 97.06 | 64.26 | 87.83 | 65.83 | 87.43 | 59.23 | 88.53 | 50.80 | 90.98 |
| | FeatureNorm (Yu et al., 2023) | 8.84 | **98.24** | 24.62 | **95.11** | 3.38 | **99.36** | 71.17 | 83.12 | 62.80 | 86.05 | 65.25 | 85.20 | 39.34 | 91.18 |
| | **MSP+SFT** | 57.15 | 91.10 | 54.68 | 91.03 | 9.00 | 98.44 | **15.29** | **97.13** | **18.27** | **96.65** | 55.41 | 89.16 | **35.05** | **93.92** |
| | **Energy+SFT** | **44.23** | **93.27** | **43.39** | 92.90 | 3.25 | 99.16 | **10.65** | 97.86 | **13.51** | 97.36 | **46.18** | 90.64 | **26.87** | **95.20** |

### 4.3 ABLATION STUDIES

**Impact of pseudo-OOD data.** We observe that if we only fine-tune the model using ID data, the performance gain would be smaller. Table 4 compares the performance of the classical `Energy` and our `SFT`. Our `SFT` achieves large improvement compared to `SFT` only using ID data, which demonstrates the effectiveness of introducing the pseudo-OOD data during fine-tuning.

Table 4: Ablation study on the impact of pseudo-OOD data during the fine-tuing on ResNetv2-101 (He et al., 2016b) and T2T-ViT-24 (Yuan et al., 2021). All values are reported in percentages.

| Model | Methods | iNaturalist | | SUN | | Places | | Textures | | Average | |
|---|---|---|---|---|---|---|---|---|---|---|---|
| | | FPR95 ($\downarrow$) | AUROC ($\uparrow$) | FPR95 ($\downarrow$) | AUROC ($\uparrow$) | FPR95 ($\downarrow$) | AUROC ($\uparrow$) | FPR95 ($\downarrow$) | AUROC ($\uparrow$) | FPR95 ($\downarrow$) | AUROC ($\uparrow$) |
| ResNetv2-101 | Energy | 64.91 | 88.48 | 65.33 | 85.32 | 73.02 | 81.37 | 80.87 | 75.79 | 71.03 | 82.74 |
| | Energy+SFT(only ID data) | 65.23 | 89.13 | 67.55 | 85.27 | 72.31 | 82.21 | 80.60 | 76.31 | 71.42 | 83.23 |
| | Energy+SFT | **34.59** | **94.21** | **52.98** | **89.23** | **63.7** | **85.04** | **30.80** | **91.94** | **45.52** | **90.11** |
| T2T-ViT-24 | Energy | 52.95 | 82.93 | 68.55 | 73.06 | 74.24 | 68.17 | 51.05 | 83.25 | 61.70 | 76.85 |
| | Energy+SFT(only ID data) | 56.03 | 83.01 | 68.53 | 73.62 | 71.11 | 70.57 | 50.11 | 84.96 | 61.45 | 78.04 |
| | Energy+SFT | **30.90** | **92.13** | **56.02** | **83.27** | **63.29** | **79.60** | **22.27** | **94.56** | **43.12** | **87.39** |

**Impact of fine-tuning steps.** Initially we would like to follow `OE` (Hendrycks et al., 2018) and fine-tune the model for several epochs. However, we observe that fine-tuning for epochs would drastically impair the OOD detection performance. Table 5 presents the performance versus the training steps. As can be seen, fine-tuning for 50 steps achieves the best performance, and further training would make the OOD detection ability gradually deteriorate. We hypothesize that this might be related to the generated pseudo-OOD data. The pseudo-OOD data could not really capture the patterns for all kinds of OOD data. Fine-tuning the model for a long time would make the model "remember" the pseudo-OOD patterns and lose the ability to recognize the real OOD data.

Table 5: Ablation study of different fine-tuning steps on ResNetv2-101 (He et al., 2016b).

| epochs or steps | pretrained model | 10 steps | 50 steps | 100 steps | 200 steps | 1 epoch | 5 epochs |
|---|---|---|---|---|---|---|---|
| FPR95 ($\downarrow$) | 71.03 | 52.27 | **45.52** | 49.78 | 53.66 | 68.5 | 73.48 |
| AUROC ($\uparrow$) | 82.74 | 88.19 | **90.1** | 88.71 | 87.32 | 84.6 | 82.90 |

**SAM-only-norm instead of SAM.** We initially used the original SAM as the optimizer but we observed that the original SAM would severely decrease the standard classification accuracy of the pre-trained model. We expect this might be related to the strong perturbation of SAM applied at every layer. Table 6 compares the performance of SAM and SAM-only-norm on different architectures. As can be seen, SAM-only-norm can simultaneously improve the OOD detection performance while maintaining the standard classification accuracy. We thus switch to SAM-only-norm in the experiments. Moreover, it would be easy to conduct theoretical analysis for SAM-only-norm as only the normalization layers are involved.

Table 6: Abalation study of SAM and SAM-only-norm fine-tuning on ResNetv2-101 (He et al., 2016b) and T2T-ViT-24 (Yuan et al., 2021). Here $\downarrow$ / $\uparrow$ means the magnitude of the accuracy would decrease / increase compared to the pre-trained model.

| Model | Optimizer | FPR95($\downarrow$) | AUROC($\uparrow$) | Accuracy (%) |
|---|---|---|---|---|
| ResNetv2-101 | SAM | 28.57 | 93.35 | 70.01 (5.19 $\downarrow$) |
| | SAM-only-norm | 45.52 | 90.11 | 75.99 (0.79 $\uparrow$) |
| T2T-ViT-24 | SAM | 93.73 | 48.38 | 0.83 (80.69 $\downarrow$) |
| | SAM-only-norm | 43.12 | 87.39 | 80.74 (0.78 $\downarrow$) |

**Impact of perturbation radius $\rho$.** One important hyper-parameter of SAM is the perturbation radius $\rho$. A large $\rho$ might result in a flat loss landscape for both ID and OOD data, while a small $\rho$ might have no impact of distinguishing OOD samples. To choose an appropriate $\rho$, we conduct an ablation and present the results in Table 7. As can be seen, $\rho = 0.5$ achieves the best performance

Table 7: Ablation study of $\rho$ different on ResNetv2-101 (He et al., 2016b).

| $\rho$ | 0.05 | 0.1 | 0.5 | 1.0 | 2.0 |
|---|---|---|---|---|---|
| FPR95 ($\downarrow$) | 46.62 | 45.81 | **45.52** | 46.02 | 47.92 |
| AUROC ($\uparrow$) | 89.74 | 89.93 | **90.11** | 89.93 | 89.31 |

Table 8: Ablation study of different optimizers on ResNetv2-101 (He et al., 2016b).

| Methods | iNaturalist | | SUN | | Places | | Textures | | Average | |
|---|---|---|---|---|---|---|---|---|---|---|
| | FPR95 ($\downarrow$) | AUROC ($\uparrow$) | FPR95 ($\downarrow$) | AUROC ($\uparrow$) | FPR95 ($\downarrow$) | AUROC ($\uparrow$) | FPR95 ($\downarrow$) | AUROC ($\uparrow$) | FPR95 ($\downarrow$) | AUROC ($\uparrow$) |
| Energy+FT(SGD) | 41.95 | 93.07 | 60.22 | 87.12 | 68.22 | 83.23 | 31.65 | 91.94 | 50.51 | 88.84 |
| Energy+FT(Adam) | 99.58 | 26.44 | 97.12 | 43.33 | 96.05 | 45.60 | 99.06 | 26.54 | 97.95 | 35.48 |
| **Energy+SFT** | **34.59** | **94.21** | **52.98** | **89.23** | **63.70** | **85.04** | **30.80** | **91.94** | **45.52** | **90.11** |
| RankFeat+FT(SGD) | 38.31 | 92.63 | 27.82 | 94.28 | **37.34** | **91.38** | 29.00 | 93.32 | 33.12 | 92.90 |
| RankFeat+FT(Adam) | 75.70 | 71.36 | 92.47 | 61.15 | 94.55 | 55.91 | 66.65 | 72.71 | 82.34 | 65.28 |
| **RankFeat+SFT** | **37.15** | **92.93** | **27.66** | **94.35** | 37.88 | 91.14 | **27.29** | **93.83** | **32.50** | **93.06** |

among different values. We note that though our SFT needs to search for an appropriate $\rho$, this process is still very efficient as the fine-tuning only takes 50 steps.

**SAM is more effective than other optimizers.** To demonstrate the superiority of the SAM optimizer, we conduct an ablation study using alternative optimizers. Table 8 compares the performance of SGD, Adam, and our SAM. Our SAM achieves better performance than the other baselines. Specifically, our SAM outperforms SGD by $4.99\%$ and $0.62\%$ in FPR95 with `Energy` and `RankFeat`, respectively. We attribute this improvement to the smooth loss landscape of ID samples optimized by SAM.

## 5 CONCLUSION

In this paper, we propose `SFT`, a real OOD-data-free and time-efficient OOD detection method by fine-tuning the pre-trained model with SAM. Our `SFT` helps existing OOD detection methods to have large performance gains across datasets and benchmarks. Extensive ablation studies are performed to illustrate different hyper-parameter choices, and theoretical analyses are presented to shed light on the working mechanism. We hope our method can bring inspiration and novel understandings to the research community of OOD detection from alternative perspectives.

## 6 LIMITATION

In this section, we honestly discuss the current limitations of our `SFT` and point out the potential directions of future work.

**More diverse and realistic pseudo-OOD data are needed.** As Fig. 2 and Table 1 indicate, our `SFT` has very large improvement on Textures and iNaturalist. This observation indicates that our pseudo-OOD data might precisely represent the image patterns of Textures and iNaturalist. However, the performance improvement is relatively limited on Places and SUN, which indicates the pseudo-OOD are not able to imitate the images of these two datasets. Moreover, in the ablation study, we show that fine-tuning for long would harm the performance. This phenomenon might be also related to the diversity of the generated pseudo-OOD data. Seeking a more powerful way to generate diverse and realistic pseudo-OOD data is an important future research direction.

**`SFT` does not suit models with BatchNorm (Ioffe & Szegedy, 2015).** We observe that our fine-tuning does not suit models with BatchNorm, such as ResNet50 (He et al., 2016a) for ImageNet-1K (Deng et al., 2009) and WideResNet (Zagoruyko & Komodakis, 2016) for CIFAR10 (Krizhevsky et al., 2009). However, our `SFT` works well on architectures with other normalization techniques (*i.e.,* LayerNorm and GroupNorm). The underlying reason why our `SFT` is not compatible with BatchNorm is worth further exploring.

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
