# Why SAM finetuning can benefit OOD Detection?
# –Appendix–

## A  Appendix

### A.1  Theorem and Proof

We extend the convergence analysis in Mueller et al. (2023) to prove our Proposition 1.

**Assumption 1.** *We assume the loss function $f : R_n \to R$ to be Lipschitz continuous gradient: there exists $L > 0$ such that*

$$\|\nabla f(v) - \nabla f(w)\|_2 \leq L\|v - w\|_2, \forall v, w \in \mathbb{R}^n. \tag{1}$$

**Assumption 2.** *There exists $M > 0$ for any sample $x_i$ ($x_i$ is id data or ood data) such that*

$$\|\nabla f_{x_i}(w)\|_2^2 \leq M, \forall w \in \mathbb{R}^n. \tag{2}$$

**Remark 1.** *If Assumption 1 holds (L-smoothness), then any $v, w \in R_n$:*

$$|f(v) - (f(w) + \nabla f(w)^T (v - w))| \leq \frac{L}{2}\|v - w\|_2^2. \tag{3}$$

Assumption 2 guarantees that the norm of the stochastic gradient is less than the upper bound of gradient norm $M$. For SAM-Only-norm, the update for the norm module is:

$$\begin{aligned} w_N^{t+1/2} &= w_N^t + \rho \frac{g_{N,x_i}(w^t)}{\|g_{N,x_i}(w^t)\|} \\ w_N^{t+1} &= w_N^t - h \, g_{N,x_i}\left(w^{t+1/2}\right) \end{aligned} \tag{4}$$

The update for the other module is:

$$\begin{aligned} w_A^{t+1/2} &= w_A^t \\ w_A^{t+1} &= w_A^t - h \, g_{A,x_i}\left(w^{t+1/2}\right). \end{aligned} \tag{5}$$

We denote the loss of the model's output as $f(w)$, the learning rate as $h$, the true gradient as $\nabla f(w^t)$ and the unbiased computational gradient as $g(w)$, assume that $h \leq 1/L$ then we have:

$$
\begin{aligned}
f(w^{t+1}) &\leq f(w^t) + \nabla f(w^t) \cdot (w^{t+1} - w^t) + \frac{L}{2}\|w^{t+1} - w^t\|^2 \\
&\leq f(w^t) - h\nabla f(w^t) \cdot g_{x_i}\left(w^{t+1/2}\right) + \frac{h^2 L}{2}\left\|g_{x_i}\left(w^{t+1/2}\right)\right\|^2 \\
&= f(w^t) - h\nabla f(w^t) \cdot g_{x_i}\left(w^{t+1/2}\right) \\
&\quad + \frac{h^2 L}{2}\left(\|\nabla f(w^t) - g_{x_i}(w^{t+1/2})\|^2 - \|\nabla f(w^t)\|^2 + 2\left(\nabla f(w^t) \cdot g_{x_i}(w^{t+1/2})\right)\right) \\
&= f(w^t) - \frac{Lh^2}{2}\|\nabla f(w^t)\|^2 + \frac{Lh^2}{2}\|\nabla f(w^t) - g_{x_i}(w^{t+1/2})\|^2 \\
&\quad - (1 - Lh)h\left(\nabla f(w^t) \cdot g_{x_i}(w^{t+1/2})\right) \\
&\leq f(w^t) - \frac{Lh^2}{2}\|\nabla f(w^t)\|^2 + Lh^2\|\nabla f(w^t) - g_{x_i}(w^t)\|^2 \\
&\quad + Lh^2\|g_{x_i}(w^t) - g_{x_i}(w^{t+1/2})\|^2 - (1 - Lh)h\left(\nabla f(w^t) \cdot g_{x_i}(w^{t+1/2})\right).
\end{aligned}
\tag{6}
$$

Take the expectation on both sides of this inequality:

$$
\begin{aligned}
\mathbb{E}[f(w^{t+1})] &\leq \mathbb{E}[f(w^t)] - \frac{Lh^2}{2}\mathbb{E}\|\nabla f(w^t)\|^2 + Lh^2 M \\
&\quad + Lh^2\|g(w^t) - g(w^{t+1/2})\|^2 - (1 - Lh)h\mathbb{E}\left[\nabla f(w^t) \cdot g(w^{t+1/2})\right].
\end{aligned}
\tag{7}
$$

For the penultimate term of Eq. 7, we have:

$$
Lh^2\|g(w^t) - g(w^{t+1/2})\|^2 \leq L^3 h^2\|w^t - w^{t+1/2}\|^2 = L^3 h^2 \rho^2.
\tag{8}
$$

For the last term of Eq. 7, we have:

$$
\begin{aligned}
\mathbb{E}\left[\nabla f(w^t) \cdot g(w^{t+1/2})\right] &= \mathbb{E}\left[\{\nabla f_N(w^t), \nabla f_A(w^t)\} \cdot \{g_N(w^{t+1/2}), g_A(w^{t+1/2})\}\right] \\
&= \mathbb{E}[\nabla f_A(w^t) \cdot (g_A(w^{t+1/2}))] + \mathbb{E}[\nabla f_N(w^t) \cdot (g_N(w^{t+1/2})) \\
&= \mathbb{E}[\nabla f_A(w^t) \cdot (g_A(w^{t+1/2}) - g_A(w^t) + g_A(w^t))] \\
&\quad + \mathbb{E}[\nabla f_N(w^t) \cdot (g_N(w^{t+1/2}) - g_N(w^t) + g_N(w^t)) \\
&= \mathbb{E}\left[\|\nabla f(w^t)\|^2\right] + \mathbb{E}[\nabla f_A(w^t) \cdot (g_A(w^{t+1/2}) - g_A(w^t))] \\
&\quad + \mathbb{E}[\nabla f_N(w^t) \cdot (g_N(w^{t+1/2}) - g_N(w^t))]
\end{aligned}
\tag{9}
$$

Using $xy \leq \frac{1}{2}\|x\|_2^2 + \frac{1}{2}\|y\|_2^2$ and the Assumption 1 leads to:

$$
\begin{aligned}
&|\mathbb{E}[\nabla f_A(w^t) \cdot (g_A(w^{t+1/2}) - g_A(w^t))] + \mathbb{E}[\nabla f_N(w^t) \cdot (g_N(w^{t+1/2}) - g_N(w^t))]| \\
&\leq \frac{1}{2}\mathbb{E}\left[\|\nabla f(w^t)\|^2\right] + \frac{L^2}{2}\|w^{t+1/2} - w^t\|^2 = \frac{1}{2}\mathbb{E}\left[\|\nabla f(w^t)\|^2\right] + \frac{L^2 \rho^2}{2}
\end{aligned}
\tag{10}
$$

Using $h \leq 1/L$ leads to:

$$
\begin{aligned}
-(1 - Lh)h\mathbb{E}\left[\nabla f(w^t) \cdot g(w^{t+1/2})\right] &\leq -(1 - Lh)h\mathbb{E}\|\nabla f(w^t)\|^2 \\
&\quad + (1 - Lh)h\left(\frac{1}{2}\mathbb{E}\|\nabla f(w^t)\|^2 + \frac{L^2 \rho^2}{2}\right)
\end{aligned}
\tag{11}
$$

Plugging Eqs. 8 and 11 into Eq. 7 gives:

$$
\begin{aligned}
\mathbb{E}[f(w^{t+1})] &\leq \mathbb{E}[f(w^t)] - \frac{Lh^2}{2}\mathbb{E}\|\nabla f(w^t)\|^2 + Lh^2\|g(w^t) - g(w^{t+1/2})\|^2 \\
&\quad - (1 - Lh)h\left(\frac{3}{2}\mathbb{E}\|\nabla f(w^t)\|^2 + \frac{L^2 \rho^2}{2}\right) \\
&\leq \mathbb{E}[f(w^t)] - \frac{h}{2}\mathbb{E}\|\nabla f(w^t)\|^2 + Lh^2 M + \frac{1}{2}hL^2\rho^2(1 + Lh)
\end{aligned}
\tag{12}
$$

Usually, we use the cross-entropy loss in multi-classification tasks. $f(w^t)$ can be thus re-written as:

$$f(w^t) = f(g(w^t)) = -log\frac{e^{max(g_i)}}{\sum\limits_{i=1}^{n} e^{g_i}} = -max(g_i) + log\sum_{i=1}^{n} e^{g_i} \tag{13}$$

where $g(w^t) = [g_1, g_2, ..., g_n]$ denotes the logit output of the model, $n$ is the total number of classes in the classification task. Thus, Eq. 12 can be rewritten as the following inequality:

$$\triangle(-max(g_i) + log\sum_{i=1}^{n} e^{g_i}) \leq -\frac{h}{2}\mathbb{E}\|\nabla f(w^t)\|^2 + Lh^2M + \frac{1}{2}hL^2\rho^2(1 + Lh) \tag{14}$$

We notice that $log\sum\limits_{i=1}^{n} e^{g_i}$ takes the same form of enengy score (Liu et al., 2020). Then, we can get the upper bound of the score change.

$$\triangle(\text{Energy Score}) \leq -\frac{h}{2}\mathbb{E}\|\nabla f(w^t)\|^2 + \triangle(max(g_i)) + Lh^2M + \frac{1}{2}hL^2\rho^2(1 + Lh) \tag{15}$$

## A.2 ADDITIONAL VISUALIZATIONS.

**Pseudo-OOD data.** Fig. 1 and 2 visualize the exemplary pseudo-OOD data generated from ID data. The pseudo-OOD data can mimic the real-world OOD data and improve the detection performance.

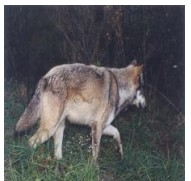 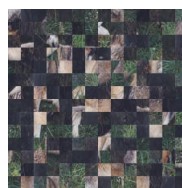 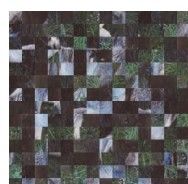

Original ID data       *Jigsaw Puzzle*       *Jigsaw Puzzle+Invert*

Figure 1: Visualization of pseudo OOD data on ImageNet-1K (Deng et al., 2009).

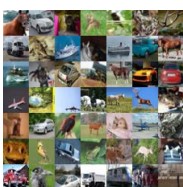 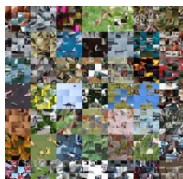 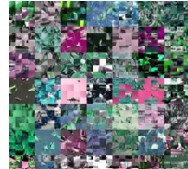

Original ID data       *Jigsaw Puzzle*       *Jigsaw Puzzle+Invert*

Figure 2: Visualization of pseudo OOD data on CIFAR-10 (Krizhevsky et al., 2009).

**Score distributions of ResNetv2-101.** Fig. 3 and 4 displays the score distributions of `MSP` and `RankFeat` with ResNetv2-101. Our `SFT` greatly reduces the overlapping of ID and OOD scores, enhancing the OOD detection performance.

**Score distributions of T2T-ViT-24.** Fig. 5, 6, and 7 depicts the score distributions of `MSP`, `Energy` and `RankFeat` with T2T-ViT-24, respectively. Empowered by our fine-tuning, the performance of baseline methods is greatly boosted to better distinguish OOD samples.

**Gradient norm and maximum output logit distributions of ResNetv2-101 on the other OOD testsets.** Fig. 8, 9 and 10 display the gradient norm of FC layer and maximum output logit distributions on iNaturalist, SUN and Places OOD testsets with ResNetv2-101. The observation from the above figures meets our previous theoretical analysis on the working mechanism of `SFT`.

**Gradient norm and maximum output logit distributions of T2T-ViT-24.** Fig. 11, 12, 13 and 14 display the gradient norm of FC layer and maximum output logit distributions on iNaturalist, SUN, Places and Textures OOD testsets with T2T-ViT-24, respectively. The observation from the above figures meets our previous theoretical analysis on the working mechanism of `SFT`: we reduce the upper bound of score change by reducing the maximum logit output.

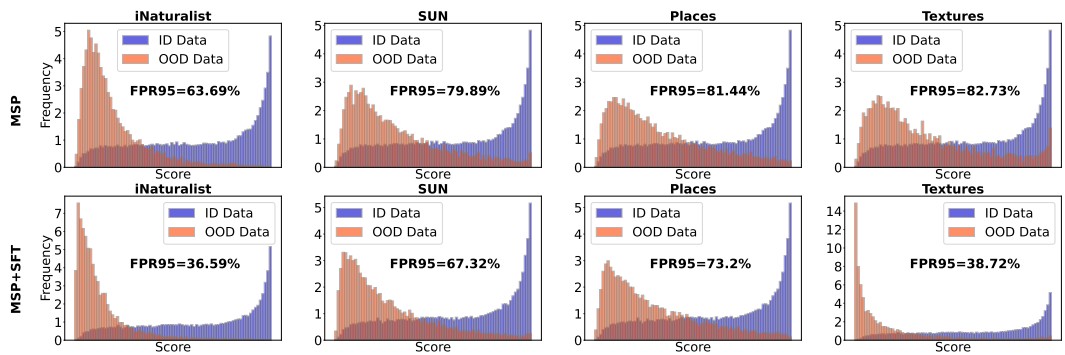

Figure 3: Score distributions of `MSP` with ResNetv2-101.

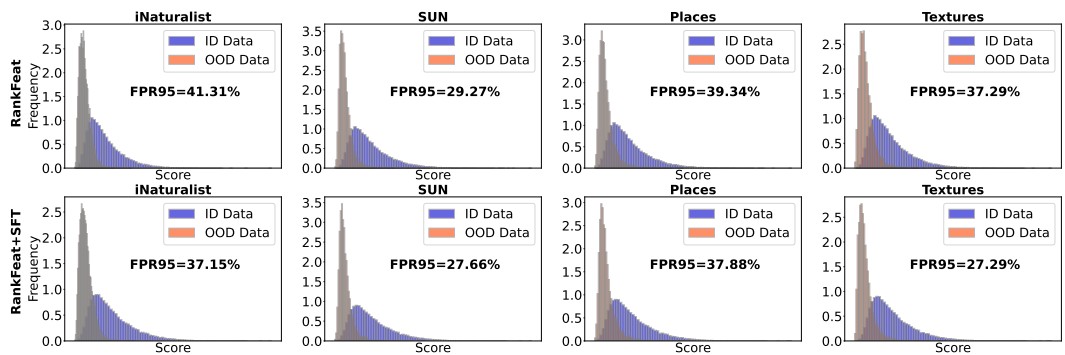

Figure 4: Score distributions of `RankFeat` with ResNetv2-101.

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

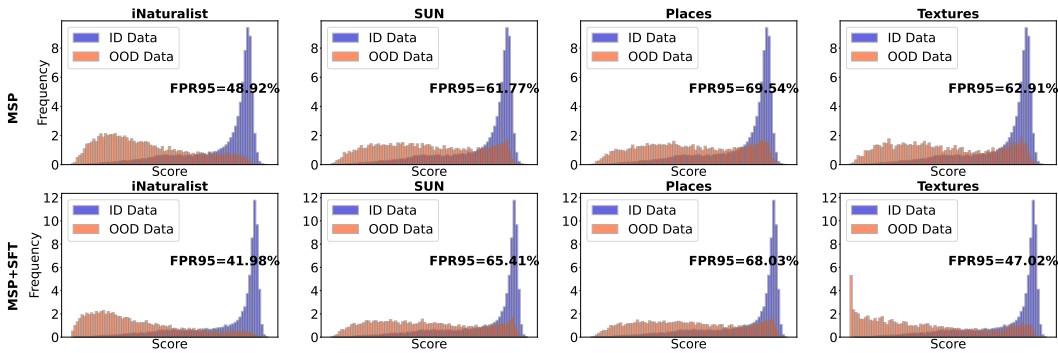

Figure 5: Score distributions of `MSP` with T2T-ViT-24.

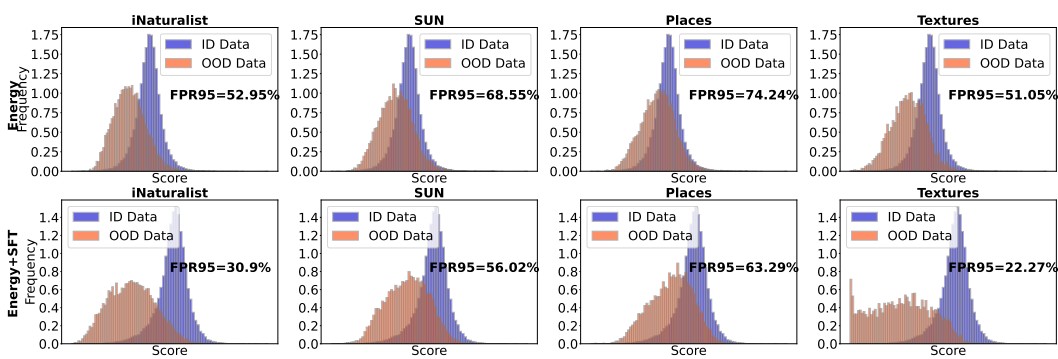

Figure 6: Score distributions of `Energy` with T2T-ViT-24.

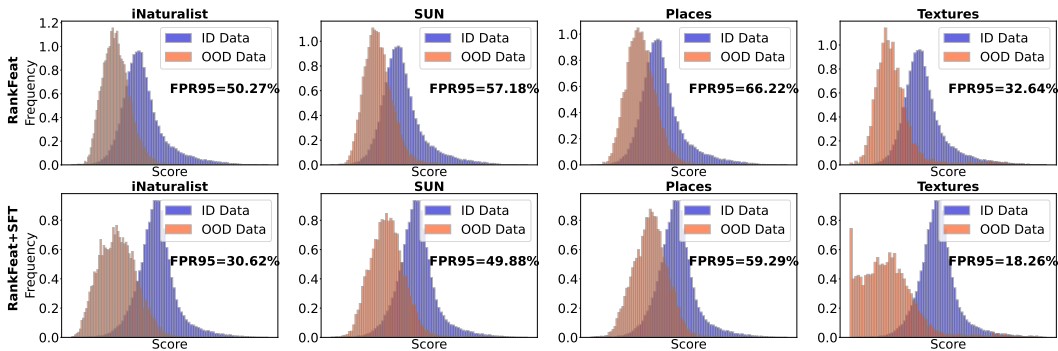

Figure 7: Score distributions of `RankFeat` with T2T-ViT-24.

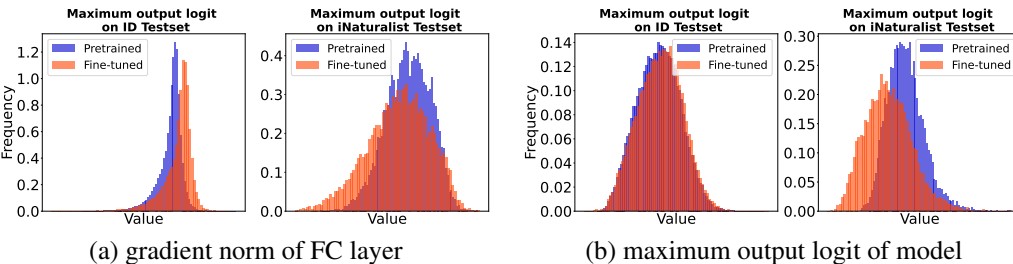

Figure 8: The gradient norm of FC layer and maximum output logit of model on ID and **iNaturalist** OOD testset with ResNetv2-101.

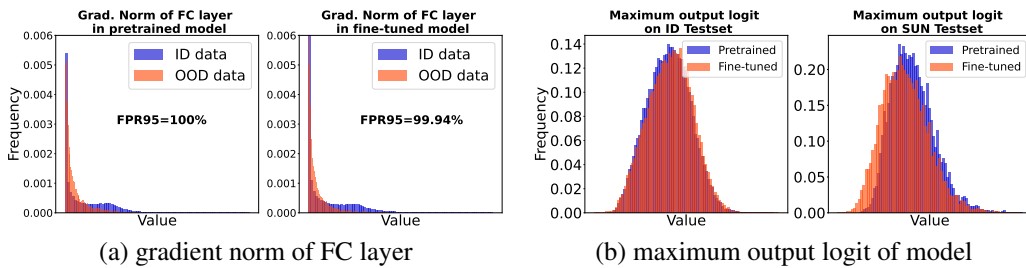

Figure 9: The gradient norm of FC layer and maximum output logit of model on ID and **SUN** OOD testset with ResNetv2-101.

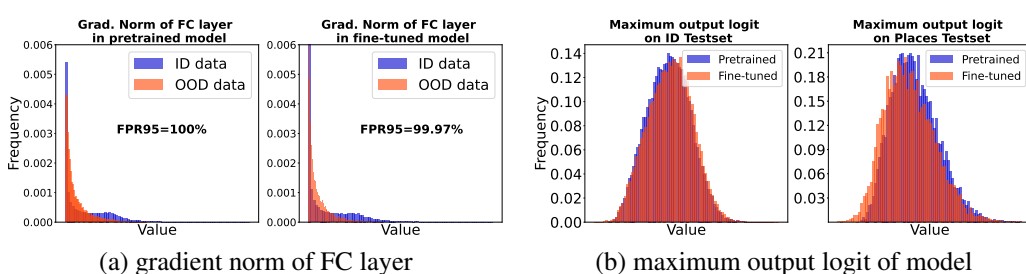

Figure 10: The gradient norm of FC layer and maximum output logit of the model on ID and **Places** OOD testset with ResNetv2-101.

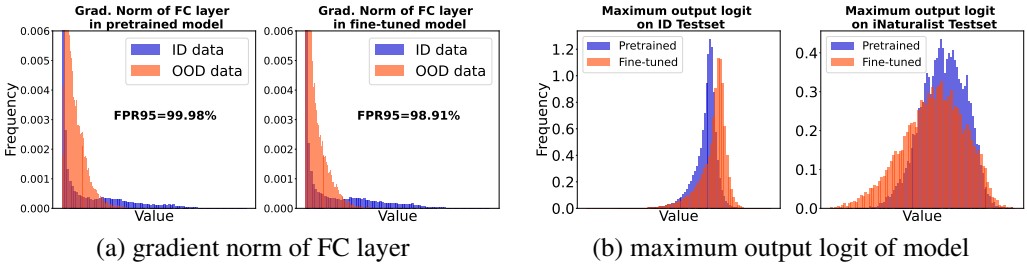

Figure 11: The gradient norm of FC layer and maximum output logit of model on ID and **iNaturalist** OOD testset with T2T-ViT-24.

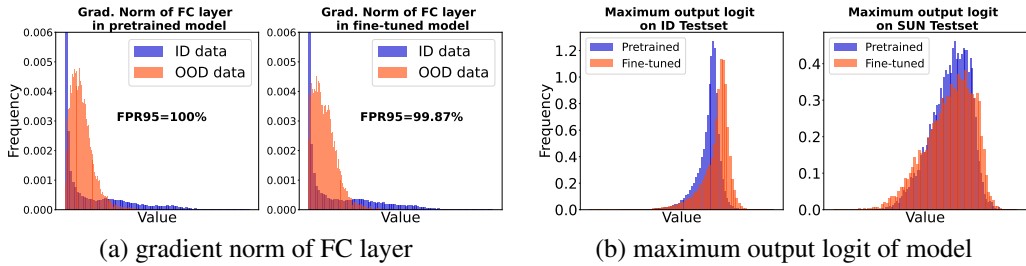

Figure 12: The gradient norm of FC layer and maximum output logit of model on ID and **SUN** OOD testset with T2T-ViT-24.

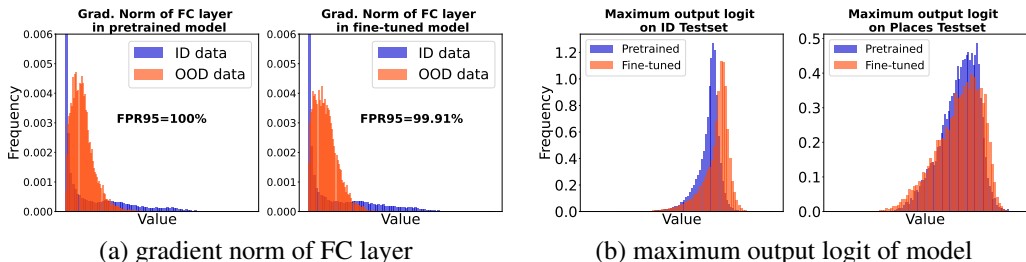

Figure 13: The gradient norm of FC layer and maximum output logit of model on ID and **Places** OOD testset with T2T-ViT-24.

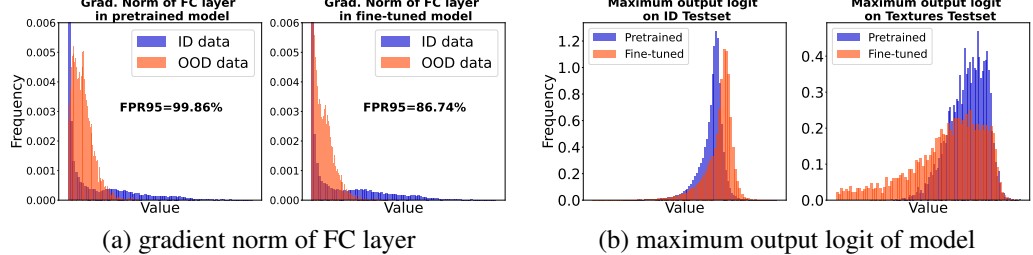

Figure 14: The gradient norm of FC layer and maximum output logit of model on ID and **Textures** OOD testset with T2T-ViT-24.