# OpenReview forum: "Why SAM finetuning can benefit Out-of-Distribution Detection?"
_ICLR.cc/2024/Conference — ICLR 2024 Conference Withdrawn Submission_

### Official Review · Reviewer_uGJJ · 2023-10-31

**Soundness:** 2 fair
**Presentation:** 3 good
**Contribution:** 2 fair
**Rating:** 5
**Confidence:** 4

**Summary:**

The paper addresses out-of-distribution (OOD) detection in machine learning, proposing a novel approach using Sharpness-aware Minimization (SAM) instead of traditional SGD for model fine-tuning. This results in improved model performance and generalization, directly benefiting OOD detection. The fine-tuning process is time-efficient, yielding significant OOD performance enhancements within just one epoch. The versatile method enhances various OOD detection strategies across different architectures, as demonstrated by state-of-the-art results on standard OOD benchmarks, with comprehensive theoretical and empirical backing provided.

**Strengths:**

1.The paper is well-written and the methodology is clear and easy to understand.
2.Extensive experiments are done to analyze various aspects of the method.
3.Leveraging SAM to improve OOD detection sounds interesting.

**Weaknesses:**

1. This article seems to solely focus on adapting SAM for the task of OOD detection, a point of innovation that appears rather weak to support the entirety of the paper.
2. The authors claim that there is no need for real OOD data; however, looking at the form of the SAM loss, it seems that utilizing real OOD data could work just as well. I am curious if employing real OOD data for fine-tuning in conjunction with SAM loss could potentially yield better results.
3. The results presented in Tables 1, 2, and 3 indicate that SFT may not always achieve optimal performance across all OOD datasets, a phenomenon that the authors have not explained in detail.
4. A number of recent related works have not been compared in this study, including DICE [1], ViM [2], ASH [3], NPOS [4], and CIDER [5].

[1] Sun, Yiyou, and Yixuan Li. "Dice: Leveraging sparsification for out-of-distribution detection." European Conference on Computer Vision. Cham: Springer Nature Switzerland, 2022.

[2] Wang, Haoqi, et al. "Vim: Out-of-distribution with virtual-logit matching." Proceedings of the IEEE/CVF conference on computer vision and pattern recognition. 2022.

[3] Djurisic, Andrija, et al. "Extremely Simple Activation Shaping for Out-of-Distribution Detection." The Eleventh International Conference on Learning Representations. 2022.

[4] Tao, Leitian, et al. "Non-parametric Outlier Synthesis." The Eleventh International Conference on Learning Representations. 2022.

[5] Ming, Yifei, et al. "How to Exploit Hyperspherical Embeddings for Out-of-Distribution Detection?." The Eleventh International Conference on Learning Representations. 2022.

**Questions:**

see weakness

---

### Official Review · Reviewer_NGyn · 2023-11-01

**Soundness:** 2 fair
**Presentation:** 2 fair
**Contribution:** 2 fair
**Rating:** 5
**Confidence:** 3

**Summary:**

This paper propose to study the OOD problem from the perspective of Sharpness-aware Minimization (SAM). Compared
with traditional optimizers such as SGD, SAM can better improve the model performance and generalization ability, and this is closely related to OOD detection.

**Strengths:**

This paper propose Sharpness-aware Fine-Tuning (SFT), a OOD data-free, and time efficient method for OOD detection by fine-tuning pre-trained model with SAM using generated pseudo-OOD data within 1 epoch. Comprehensive experimental results indicate that SFT brings improvements in various OOD detection benchmarks and methods.

**Weaknesses:**

1. this paper doesn't have enough novelty.
2. The compared methods mostly are before 2022.
3. This paper lack enough reference, like "Out-of-distribution detection with an adaptive likelihood ratio on informative hierarchical VAE"

**Questions:**

The compared methods mostly are before 2022. Is there any newer methods to fairly compare?

---

### Official Review · Reviewer_eCTD · 2023-11-02

**Soundness:** 3 good
**Presentation:** 3 good
**Contribution:** 3 good
**Rating:** 6
**Confidence:** 4

**Summary:**

This paper presents a novel technique, Sharpness-aware Fine-Tuning (SFT), which enhances the post hoc methods for out-of-distribution (OOD) detection. By applying SFT to various post hoc methods, the authors achieve remarkable performance gains on different benchmarks. The paper assesses the efficacy of SFT on two benchmark datasets, SFT elevates the baseline performance of MSP, Energy, and RankFeat methods in terms of average FPR95 and AUROC. Likewise, for VGG11 on CIFAR-10, SFT improves the Energy baseline and propels the performance of MSP to a competitive level. The authors also explore the influence of different normalization layers on the performance of SFT. They discover that SFT is consistent in augmenting the performance across networks with different normalization layers. Overall, the paper’s contributions encompass the introduction of the SFT technique, which boosts the performance of post hoc methods for OOD detection, and the illustration of its effectiveness across different benchmarks and normalization layers.

**Strengths:**

-	The paper presents a novel method dubbed Sharpness-aware Fine-Tuning (SFT) that enhances the detection of out-of-distribution (OOD) samples. The authors devise a fine-tuning scheme that employs the SAM optimizer and pseudo-OOD data crafted by Jigsaw Puzzle Patch Shuffling and RGB Channel Shuffling. They show that SFT markedly boosts the OOD detection performance of baseline methods across diverse benchmarks.
-	The authors devise an innovative approach that exploits the sharpness of the loss landscape to augment OOD detection. The incorporation of SAM and pseudo-OOD data in the fine-tuning procedure is a clever synthesis of existing techniques.
-	The paper offers extensive experimental results and ablation studies to corroborate the efficacy of SFT. The theoretical analysis elucidates the underlying mechanism of SFT.
-	Overall, the paper contributes to the field of OOD detection by proposing a simple and effective method that attains significant performance gains.

**Weaknesses:**

- The paper does not provide a comparison with OE methods that rely on OOD data for training. Although SFT does not introduce external OOD data, it generates partial OOD data, and both SFT and OE methods belong to the category of incorporating OOD data into training. Therefore, a comparison with the OE method is indispensable. It is not mandatory to surpass the OE method, but the efficacy of the SFT method can be demonstrated.
- The main contribution of the paper stems from the generation of pseudo OOD data, which has achieved significant performance improvement. However, it is still necessary to elucidate the training and computational costs associated with SFT compared to other methods.

**Questions:**

- Although this paper focuses on OOD detection tasks, does SFT have an impact on the performance of IDs? Will it result in a decline in the classification performance of IDs?
- The paper mentions that SFT cannot use the BatchNorm model, which may be due to substantial differences in the distribution of generated OOD data and ID data. Can we use the OOD independent BatchNorm to address this issue?

---

### Official Review · Reviewer_4Xsz · 2023-11-08

**Soundness:** 1 poor
**Presentation:** 1 poor
**Contribution:** 1 poor
**Rating:** 3
**Confidence:** 5

**Summary:**

The paper introduces a lightweight method called Sharpness-aware Fine-Tuning (SFT) for improving Out-of-Distribution (OOD) detection in machine learning models. It leverages Sharpness-aware Minimization (SAM) for fine-tuning pre-trained models and appears to enhance OOD detection performance across various benchmarks and methods. The method is practical as it doesn't require real OOD data for training and demonstrates robustness across different datasets and architectures.

**Strengths:**

- The proposed method is lightweight and doesn't rely on true OOD data for training, making it practical and cost-effective.
- The paper provides clear and insightful illustrations.
- Experiments are conducted using different backbone architectures and on two different datasets, ImageNet and CIFAR-10.
- Multiple ablation studies are conducted.

**Weaknesses:**

- The paper's "Related Work" section could provide more context and insights into how the proposed method relates to existing research. In particular, except for the OOD detection with discriminative models paragraph, all paragraphs are written as a catalog of related papers. Moreover, the link between generative models for OOD detection and the proposed method is unclear.
- The paper doesn't provide a clear distinction between the improvements from Outlier Exposure fine-tuning and the optimization benefits of Sharpness-aware Minimization. This point is critical as OE is a strong baseline and might be the main source for OOD detection improvement.
- Some parts of the paper lack clarity and are hard to follow. In particular, is Proposition 1 related to the upper bound of the Energy score or the upper bound of the variation of the Energy score? Moreover, the link between the bound and the better separability is not clear in the main paper and should be made explicit.
- The method appears to struggle in near-OOD scenarios, as seen in the results for the iNaturalist dataset, where it falls significantly below top-performing baselines. I guess this behavior is expected due to the proximity of near-OOD samples with ID ones thus limiting the impact of the Sharpness-aware Fine-Tuning.

**Questions:**

- Why are KL Matching and MOS not evaluated in Table 2 ? Why are DICE and FeatureNorm only evaluated on CIFAR-10?
- The choice of backbones for OOD detection experiments is non-standard as most papers used Resnet, WideResnet, or ViT-like backbone trained on the ID data. Here, why use big transfer for conducting the evaluation, and are all the baselines evaluated with the same backbone in Table 1 ? This point is critical as different backbones show drastically different OOD detection performances. In Tables 2 and 3, it seems that all the experiments are conducted with the same backbones. However, the choice of T2T-ViT and VGG-11 rather than a simple Vit or a ResNet is unclear and makes comparisons between OOD papers difficult.
- In the appendix, Remark 1 only holds if $f$ is convex which is hardly the case for the deep neural networks. The following derivation starting from eq. (6) might thus be incorrect in the general setting.

---

### Official Review · Reviewer_F3yd · 2023-11-09

**Soundness:** 1 poor
**Presentation:** 1 poor
**Contribution:** 3 good
**Rating:** 3
**Confidence:** 3

**Summary:**

The paper argues that existing post-hoc energy-based OOD detection methods can be improved by fine-tuning with SAM on synthetic OOD data. The paper provides theoretical and empirical justification for the method.

**Strengths:**

The core contributions of the paper are important and the empirical validation (except for the issues raised below) seems to be complete. The authors thoroughly compare with prior algorithms and evaluate on a wide variety of datasets. They perform extensive ablations.

**Weaknesses:**

Writing:

The paper is missing important definitions and descriptions of relevant algorithms and terms, and generally lacks the necessary precision to be able to fully understand the paper. This includes: SAM, energy, the "statistical distance" $D$ (from equation 2), the precise definition of $\mathcal{D}_\text{out}$, summaries of the relevant algorithms. Even when provided with a citation that includes the relevant definitions, I think it would be very helpful to provide at least a brief summary of each term and to clarify terminology.

In addition, at times, the notation is a bit confusing: $f$ is used as both a neural network and a loss function. In equation 3, the parameters are written as $w^t$ but $t$ is never used.

Theory:

The statement of Proposition 1 is lacking some formality. As mentioned above, "Energy" should be properly defined. In addition, it was unclear to me (before reading the proof) what the authors mean by "the change during the fine-tuning process", how $\Delta$ is defined, and what the authors mean by "an irreducible constant". In addition, the comment that "the upper bound of the change of Energy score can improve the separation between ID and OOD" should not be included in the statement of the proposition. In addition, the fine-tuning process should be clearly and precisely defined.

The statement is also only proved for a single step of fine-tuning, and not for the full fine-tuning process.

The proof (and statement) of Proposition 1 is extremely similar to Theorem C.5 in Mueller et al. (2023), and the majority of the proof is copied verbatim. I think it would help to clearly delineate the contribution of this paper and that of prior work and cite the relationship in the main paper.

I'm also not sure I understand the purpose of the Proposition: the bound establishes a relationship between $\Delta(\operatorname{Energy})$ and $\Delta(\operatorname{max}(g))$. As I understand, the authors argue that since $\Delta(\operatorname{max}(g))$ is empirically verified to be smaller for OOD points than ID points, and thus $\Delta(\operatorname{Energy})$ will also be smaller. However, this does not follow from establishing only an upper bound, and secondly, why not directly verify that $\Delta(\operatorname{Energy})$ is smaller, and skip the comparison with $\Delta(\operatorname{max}(g))$ (which the authors do in Figure 4)?

Empirical:

In order to have a fair comparison, it's necessary to tune hyperparameters (learning rate, batch size, weight decay, and $\rho$) which are likely to have a potentially large impact on the model, and may need to be tuned differently for different models, datasets, and algorithms. (It's worth noting that the authors do include an ablation where they vary $\rho$ for one architecture). There are a couple potential issues with the results (see questions section below), that cast doubt for me as to whether the results are sound.


References:

Maximilian Mueller, Tiffany Vlaar, David Rolnick, and Matthias Hein. Normalization layers are all
that sharpness-aware minimization needs. arXiv preprint arXiv:2306.04226, 2023.

**Questions:**

There are a couple empirical results that appear out of the ordinary to me, and I was wondering if the authors could comment:
- In Table 6, the accuracy for ViT with SAM is near 0, a drop of 80% compared to the pre-training model. This, to me, suggests that SAM is likely not properly tuned, e.g., $\rho$ is too large and thus the training is unstable.
- In Table 8, Adam seems to do exceptionally worse than SGD and SFT. Typically Adam expects a smaller learning rate than SGD, could this cause training instabilities that lead to such low accuracy?